# Plasma Troponins Identify Patients with Very Low-Risk Acute Pulmonary Embolism

**DOI:** 10.3390/jcm12041276

**Published:** 2023-02-06

**Authors:** Bartosz Karolak, Michał Ciurzyński, Marta Skowrońska, Katarzyna Kurnicka, Magdalena Pływaczewska, Aleksandra Furdyna, Katarzyna Perzanowska-Brzeszkiewicz, Barbara Lichodziejewska, Szymon Pacho, Michał Machowski, Piotr Bienias, Małgorzata Wiśniewska, Marek Gołębiowski, Piotr Pruszczyk

**Affiliations:** 1Department of Internal Medicine & Cardiology, Medical University of Warsaw, Lindleya 4 St., 02-005 Warsaw, Poland; 2Department of Pulmonary Circulation, Thromboembolic Diseases and Cardiology, European Health Center, Centre of Postgraduate Medical Education, Borowa 14/18 St., 05-400 Otwock, Poland; 3Department of Clinical Radiology, Medical University of Warsaw, Chałubińskiego 5 St., 02-004 Warsaw, Poland

**Keywords:** pulmonary embolism, risk stratification, troponin, echocardiography

## Abstract

Introduction: Although in the non-vitamin K oral anticoagulants (NOAC) era majority of low-risk acute pulmonary embolism (APE) patients can be treated at home, identifying those at very low risk of clinical deterioration may be challenging. We aimed to propose the risk stratification algorithm in sPESI 0 point APE patients, allowing them to select candidates for safe outpatient treatment. Materials and methods: Post hoc analysis of a prospective study of 1151 normotensive patients with at least segmental APE. In the final analysis, we included 409 sPESI 0 point patients. Cardiac troponin assessment and echocardiographic examination were performed immediately after admission. Right ventricular dysfunction was defined as the right ventricle/left ventricle ratio (RV/LV) > 1.0. The clinical endpoint (CE) included APE-related mortality and/or rescue thrombolysis and/or immediate surgical embolectomy in patients with clinical deterioration. Results: CE occurred in four patients who had higher serum troponin levels than subjects with a favorable clinical course (troponin/ULN: 7.8 (6.4–9.4) vs. 0.2 (0–1.36) *p* = 0.000). Receiver operating characteristic (ROC) analysis showed that the area under the curve for troponin in the prediction of CE was 0.908 (95% CI 0.831–0.984; *p* < 0.001). We defined the cut-off value of troponin at >1.7 ULN with 100% PPV for CE. In univariate and multivariate analysis, elevated serum troponin level was associated with an increased risk of CE, whereas RV/LV > 1.0 was not. Conclusions: Solely clinical risk assessment in APE is insufficient, and patients with sPESI 0 points require further assessment based on myocardial damage biomarkers. Patients with troponin levels not exceeding 1.7 ULN constitute the group of “very low risk” with a good prognosis.

## 1. Introduction

Acute pulmonary embolism (APE) is one of the most frequent causes of cardiovascular mortality worldwide [1]. Numerous predisposing genetic and environmental factors are known, such as lower limb fracture and joint replacement, major trauma, surgery, other causes of immobility, neoplastic diseases, estrogen use, pregnancy, and others [1]. Moreover, COVID-19 is a new, strong risk factor for APE, which is likely to increase the frequency of APE in the general population [2,3].

Myocardial damage caused by APE depends on the percentage of the pulmonary arterial bed involved. Occlusion of a single segmental pulmonary artery may remain asymptomatic and clinically inconsequential. On the other hand, obstruction of pulmonary arteries by large thrombi may result in significant growth of pulmonary vessel resistance, which triggers compensatory mechanisms increasing heart rate and right ventricular (RV) contractibility in order to maintain flow through occluded vessels. As a consequence, RV oxygen demand rises, which, in case of inadequate supply, may damage the cardiomyocytes. RV distress may be detected using imaging studies (e.g., echocardiography, computed tomography (CT)), whereas elevated serum troponin level demonstrates myocardial injury. Another biomarker specific to APE mentioned in the guidelines is d-dimers. However, its clinical use is currently limited to diagnostic processes rather than risk stratification [1].

The clinical spectrum of APE is broad; therefore, risk stratification of early death is mandatory to establish an appropriate management strategy. Current guidelines of the European Society of Cardiology (ESC) propose a stepwise risk stratification approach, using a combination of clinical findings, imaging, and biochemical markers, to distinguish between patients with high, intermediate, and low risk of an early adverse outcome [1]. Hemodynamic instability is a rare form of clinical manifestation, as it indicates central and/or extensive APE with markedly reduced hemodynamic reserve. Mortality in this group exceeds 15%, increasing to 65% when associated with cardiac arrest [4]; therefore, such patients require prompt diagnosis and reperfusion treatment [5].

The vast majority of APE patients are hemodynamically stable at hospital admission; however, the risk of further clinical deterioration can not be neglected. Thus, early risk assessment is the key to successful APE management. One of the most challenging tasks is to identify, within the large group of normotensive and apparently stable patients, those at “very low” risk to permit early discharge and ambulatory treatment [6,7]. Avoiding unnecessary hospitalization reduces treatment costs, improves the quality of patients’ lives, and reduces the burden on healthcare during the COVID-19 pandemic [8,9]. In the era of vitamin K antagonists (VKAs), ambulatory treatment of APE patients was difficult. However, according to the current guidelines, non-vitamin K antagonist oral anticoagulants (NOAC) nowadays are the standard of care for the treatment of APE [1]. The factor Xa inhibitors (apixaban and rivaroxaban) can be administered as a single oral drug regimen from the first day of therapy, obviating the need for initial parenteral treatment with low-molecular-weight heparin (LMWH). Thus, the majority of “very low” risk patients can be treated with factor Xa inhibitor at home [6,10,11]. There is evidence that the classification of APE patients into the low-risk category solely on the basis of clinical parameters is insufficient, and RV function and injury should be assessed (imaging and biomarkers approach) [1,6,12]. Therefore, we aimed to propose the risk stratification algorithm in sPESI 0 point APE patients, allowing them to identify those with a “very low” risk of clinical deterioration which can be safely treated in ambulatory care.

## 2. Materials and Methods

This was an analysis of the ongoing prospective observational study “PE-Aware”, registered at ClinicalTrials.gov (unique identifier NCT03916302). This study was approved by the local ethics committee. The ethics approval number is KB 88/2008.

### 2.1. Study Population

We analyzed data of 1151 adults (517 males (M), 634 females (F), age median = 67 (51;79)), consecutive APE patients, managed in a single referral center during the period January 2006–December 2019. All patients were hemodynamically stable at admission, with systolic blood pressure ≥ 90 mmHg and without signs of peripheral hypoperfusion. Diagnosis of APE was made when thromboemboli were visualized in at least one segmental pulmonary artery in computed tomography pulmonary angiography (CTPA) and when symptoms of pulmonary embolism had been present for no longer than 14 days before the diagnosis.

According to the ESC guidelines, low-risk APE was diagnosed when the simplified pulmonary embolism severity index (sPESI) was 0 points, and RV dysfunction and myocardial injury markers (if assessed) were negative [1]. In the remaining patients, further risk assessment was performed. Patients with both evidence of RV dysfunction (by echocardiography or CTPA) and elevated troponin levels were classified into the intermediate-high-risk category. Patients in whom RV was normal at imaging tests and/or had normal cardiac biomarkers levels formed the intermediate–low-risk group [1]. Patients diagnosed with chronic thromboembolic pulmonary hypertension and participants in therapeutic clinical trials were not included in this study.

### 2.2. Echocardiography

Echocardiographic examination was performed with a Philips iE 33 system (Philips Medical System, Santa Fe Springs, CA, USA) with 2.5–3.5 MHz transducers within the first 24 h after admission, and results were digitally recorded. Patients were examined in the left lateral position. The dimensions of the right and left ventricles were measured in the four-chamber RV-focused view at the level of the mitral and tricuspid valve tips in late diastole, as defined by the R wave of the continuous ECG tracing. Right ventricular dysfunction was defined as the right ventricle/left ventricle ratio (RV/LV) in the apical four-chamber view > 1.0. Left ventricular ejection fraction (LVEF) was calculated according to Simpson’s formula, employing a two-dimensional image of the LV chamber during systole and diastole in the four- and two-chamber apical views [13]. The agreement analysis showed high interclass and intraclass correlations for echocardiographic assessment in our center and was published previously [14].

### 2.3. Biochemical Analysis

Blood samples were collected from patients within the first 24 h after admission. Serum cardiac troponin I (cTnI) and high-sensitivity cardiac troponin T (cTnT-hs) were measured quantitatively using an automated sandwich electrochemiluminescence immunoassay (Roche Diagnostics GmbH, Mannheim, Germany). Levels above 0.014 ng/mL for cTnT-hs and 0.1 ng/mL for cTnI were considered elevated. Plasma D-dimer concentrations were measured using an automated enzyme-linked fluorescent assay VIDAS D-Dimer Exclusion (bioMerieux, Marcy-l’Étoile, France) with a threshold of 500 ng/mL in diagnosing APE.

### 2.4. The Clinical Endpoint (CE) of the Study

The clinical endpoint (CE) of the study was defined as a combination of in-hospital, APE-related mortality and/or rescue thrombolysis and/or immediate surgical embolectomy in patients with hemodynamic deterioration, which was defined by the occurrence of at least 1 of the following: (1) need for cardiopulmonary resuscitation; (2) systolic blood pressure below 90 mm Hg for at least 15 min with signs of end-organ hypoperfusion; or (3) need for intravenous catecholamines in vasopressor doses. 

### 2.5. Statistical Analysis

The normal distribution of the analyzed parameters was verified with the Shapiro–Wilk test. Parameters with non-normal distribution are expressed as medians followed by interquartile ranges. For the comparison of parameters between two groups with such distribution, the Mann–Whitney U test was used. Fisher’s exact *t*-test was used for comparisons of categorical variables. Receiver-operating curves (ROC) were used for the assessment of optimal cut-off values for the prediction of the clinical endpoint. Diagnostic performance markers (sensitivity and specificity) were calculated for the chosen cut-off values. Comparison between areas under the curve (AUC) was made pairwise using the method described by DeLong et al. [15]. Multiple logistic regression analysis was used to identify independent predictors of the CE in the analyzed group of patients for whom full echocardiographic data were available.

## 3. Results

The study group consists of 1151 consecutive normotensive patients with APE. Intermediate-risk APE was diagnosed in 741 patients, and low-risk APE in 410 patients. One patient was excluded from the final analysis due to coexisting acute myocarditis, confirmed in cardiac MRI. Initially, all patients received body mass-adjusted low molecular-weight heparin, activated partial thromboplastin time-adjusted unfractionated heparin intravenously, rivaroxaban 15 mg b.i.d. or apixaban 10 mg b.i.d. The flow of patients is presented in Figure 1.

In the final analysis, we included 409 sPESI 0 point patients (210 M, 199 F, age median = 52 years (39; 66)). In this group, rescue thrombolysis was performed due to hemodynamic collapse in 3 (0.73%) patients, 2 of them survived. One patient needed immediate surgical embolectomy. In-hospital APE-related mortality was 0.24% (1 patient), and all-cause mortality was 0.48% (2 patients, 1 patient died due to intracranial hemorrhage). The clinical endpoint (CE), which included APE-related death (1 subject, 0.24%) and/or thrombolysis (3 subjects, 0.73%), and/or surgical embolectomy (1 subject, 0.24%) occurred in 4 (0.98%) patients. Clinical characteristics of APE subjects are provided in Table 1.

Patients who experienced CEs had higher serum troponin levels than subjects with a favorable clinical course. Moreover, right ventricular dysfunction (RVD) at echocardiography was more frequent in patients with CEs. No statistically significant difference in d-dimer level between the studied groups was noted. ROC analysis showed that the area under the curve for troponin in the prediction of CE was 0.908 (95% CI 0.831–0.984), *p* < 0.001 (Table 2). We defined the cut-off value of troponin at >1.7 ULN with 100% PPV for CE. Therefore, all patients with CE (4; 0.98%) had troponin above this value. Importantly, 322 patients (78.7%) of the sPESI 0 group had cTnT concentrations below 1.7 ULN. The AUC in the prediction of CE for RV/LV > 1 was 0.725 (95% CI 0.417–0.999), *p* = 0.15 (Table 2, Figure 2).

In univariate analysis, elevated serum troponin levels were associated with an increased risk of CE, whereas RV/LV > 1.0 was not (Table 3).

In the multivariable analysis, we obtained similar results. Elevated serum troponin was associated with an increased risk of CE. We did not find statistical significance in RV/LV (Table 4).

We proposed a simple, commonly available risk stratification algorithm to identify “very low”-risk APE patients (Figure 3).

## 4. Discussion

The main finding from our study is that solely clinical risk assessment in APE is insufficient, and patients with sPESI 0 points require further assessment based on myocardial damage biomarkers. Outpatient treatment is safe when the serum troponin level does not exceed 1.7 ULN. A total of 78.7% of patients in our cohort could have been classified as a “very low” risk group with an excellent prognosis, whereas in the remaining 22.3% of patients, the CE event rate was 4.6%. Our results may have important clinical implications for the acute phase treatment of patients who appear to be at low risk of early death or severe complications based on clinical assessment alone. The clinical manifestation of APE may vary between each individual: in subjects with cardiogenic shock, the mortality rate exceeds 50%, while in hemodynamically stable patients, the prognosis is good, with a mortality rate of <1% [1]. Fortunately, about 40% of APE patients are at low risk with a good prognosis, and the majority of them may be safely discharged from the hospital and treated at home [1,16]. However, it is worth noting that some patients, who appear to be at low risk of early death according to the clinical criteria alone, may deteriorate and require reperfusion treatment. Therefore, exclusive clinical assessment is insufficient, and complete risk stratification should include the evaluation of RV function.

Although there are several scales, which were well validated in APE patients (e.g., PESI, sPESI, Hestia), the ease of use should also be considered, suggesting that sPESI may be best suited in everyday clinical practice. The clinical value of the PESI and sPESI scales is well established. Their most important strength lies in the reliable identification of patients at low risk of early mortality [16,17]. In a paper published by Aujesky D. et al., 344 low-risk APE patients (PESI I and II class) were enrolled. They were randomized to initial outpatient or inpatient treatment with enoxaparin (≥5 days) followed by oral anticoagulation. Outpatient treatment appeared to be non-inferior when compared with standard inpatient treatment [18]. However, the use of clinical scores does not consider the functional status of the right ventricle, which has repeatedly been shown to be a key determinant of APE prognosis [19,20].

Recent ESC guidelines proposed that assessment of the RV function by imaging methods or laboratory biomarkers should be considered, even in the presence of a low PESI or a negative sPESI [1]. In our paper, 4 out of 409 APE patients with sPESI 0 points hemodynamically deteriorated, which suggests that exclusive risk assessment based on clinical scales is insufficient.

Our results are consistent with the previous ones indicating that low-risk patients are at higher risk of early mortality if RV dysfunction is present. Barco et al. published a meta-analysis of 21 studies, including 3295 APE low-risk patients. Early all-cause mortality rates in patients with vs. without troponin elevation was 3.8% (95% CI 2.1–6.8%) vs. 0.5% (95% CI, 0.2–1.3%) [12]. In another meta-analysis published recently by Beccatini et al., data from 5010 low-risk patients from 18 studies were pooled. Elevated troponin was associated with short-term death (OR 2.78, 95% CI 1.06–7.26) and death within 3 months (OR 3.68, 95% CI 1.75–7.74) [21]. Subgroup analysis for low-risk APE in the meta-analysis by El Menyar et al. published in 2019 demonstrated a connection between elevated serum troponin level and mortality for high-sensitivity (OR 6.93, 95% CI 1.34–35.78) and all troponin assays (OR 9.02, 95% CI 2.51–32.35) [22]. In the study by Hakemi et al., enrolling 298 APE low-risk patients, selected according to PESI, no deaths occurred in the troponin-negative group vs. nine (6%) in the troponin-positive group. Moreover, no hard events, defined as in-hospital death, CPR, or thrombolytic therapy, were observed in the troponin-negative group vs. 15 (9%) in the hsTnT-positive group (*p* < 0.001) [23]. Taking available scientific data as well as our results into consideration, we suggest cTnT-hs assessment in each APE patient, despite low risk according to the sPESI scale. 

Numerous studies examined the usefulness of D-dimer assessment in predicting outcomes in APE; however, the results are equivocal. Meta-analysis of 22 studies by Beccattini et al. suggested an association between D-dimer level and short-term and 3-month mortality in hemodynamically stable patients with APE. Nevertheless, the investigators were not able to establish an optimum prognostic cut-off value [24]. Another study by Song et al. identified D-dimer as an independent predictor of in-hospital mortality (OR = 1.07; 95% CI, 1.003–1.143; *p* = 0.041). However, this study did not involve exclusively hemodynamically stable patients [25]. On the other hand, in the study by Stein et al., involving 292 normotensive patients, D-dimer level was not associated with increased in-hospital PE-related and all-cause mortality [26]. Current ESC guidelines do not recognize D-dimer assessment as a valuable risk stratification tool; the use of it remains limited to the diagnostic process [1]. Taking available scientific data into consideration, it is possible that D-dimer might play a role in outcome prediction; however, as it may be affected by several factors (e.g., inflammatory process, pregnancy, malignancy), its use as a single predictor does not seem adequate.

High-sensitivity troponin assessment is a simple and not expensive test, available in most hospitals at any time, whereas RV echocardiographic assessment is not always immediately reachable, and the interpretation depends on the experience of the physician. There are many echocardiographic parameters used for risk stratification in APE patients. Of these, RV/LV diameter ratio > 1.0 and a TAPSE < 16 mm are findings for which an association with unfavorable prognosis has most frequently been reported [1,27]. Echocardiography is a useful tool for risk assessment but with some limitations mentioned above.

Recent ESC guidelines recommended the use of computed tomography pulmonary angiography (CTPA) for risk stratification [1]. However, the prognosis of CTPA-assessed RV dilatation is unclear in patients with an sPESI of 0. Jimenez D. et al. published PROTECT study that included 848 normotensive patients with APE. Unfortunately, no association of CTPA-assessed RV dilatation with 30-day all-cause mortality was observed [28]. Similarly, Cote B. et al. reported that among 779 patients with sPESI 0, 420 (54%) and 299 (38%) had RV/LV ≥ 0.9 and ≥1.0, respectively. No difference in the primary outcome was observed, 0.95% (95% CI 0.31–2.59) versus 0.56% (95% CI 0.10–2.22; *p* = 0.692) and 1.34% (95% CI 0.43–3.62) versus 0.42% (95% CI 0.07–1.67; *p* = 0.211) with RV/LV ≥ 0.9 and ≥1.0, respectively [29]. In contrast, two meta-analyses revealed an increased risk of death in patients with RV dysfunction by CTPA [30,31]. Undoubtedly, CTPA plays an important role in the integrated algorithm of risk assessment. Nevertheless, further investigations are needed to evaluate the usefulness of this method in sPESI 0 patients.

### Study Limitation

This was a single-center study; therefore, our conclusions should be interpreted with caution. Moreover, due to the low-risk characteristics of studied patients, there were only a few patients reaching the clinical endpoints. In addition, causes of death were not externally adjudicated. Therefore, our findings need validation in an external cohort of APE patients, especially the usefulness of the cut-off value of 1.7 ULN for troponin measurements. Although our study did not confirm that RV/LV diameter ratio > 1.0 predicts adverse events in the low-risk group, the number of endpoints is too low to exclude echocardiography from the risk stratification process in those patients.

## 5. Conclusions

Solely clinical risk assessment in hemodynamically stable APE patients is insufficient, and all subjects with sPESI 0 points require further risk stratification based on myocardial damage biomarkers. Patients with troponin not exceeding 1.7 ULN constitute the group of “very low risk” with a good prognosis. In such patients, early oral anticoagulation and outpatient treatment should be considered. This can improve the quality of life and reduce the costs of treatment and the burden on hospitals during the COVID-19 pandemic.

## Figures and Tables

**Figure 1 jcm-12-01276-f001:**
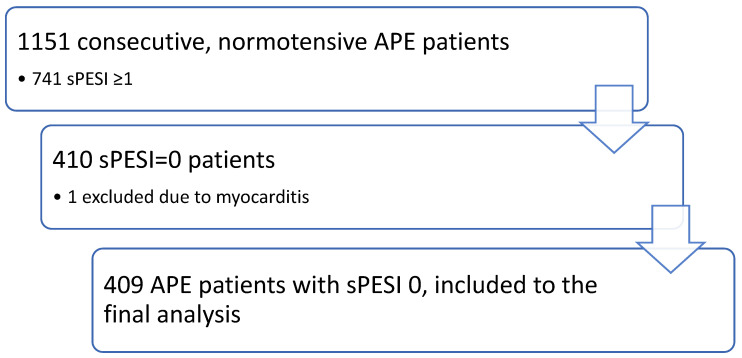
Flow of patients.

**Figure 2 jcm-12-01276-f002:**
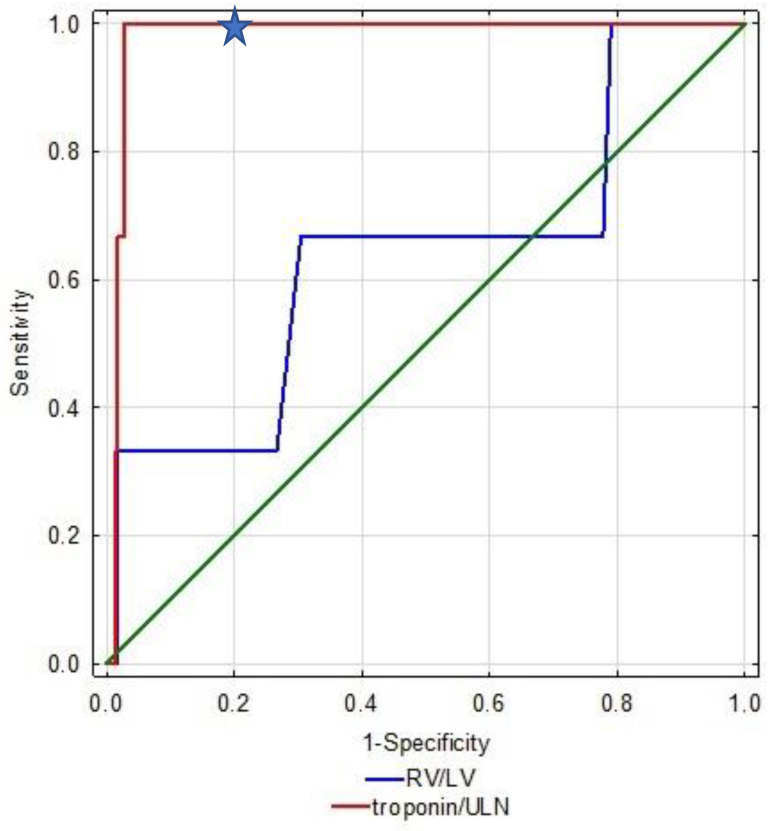
ROC curves for sPESI = 0 pts with full data on troponin levels and RV > LV ratio. ULN—upper limit of normal. The proposed cut-off is marked with a star.

**Figure 3 jcm-12-01276-f003:**
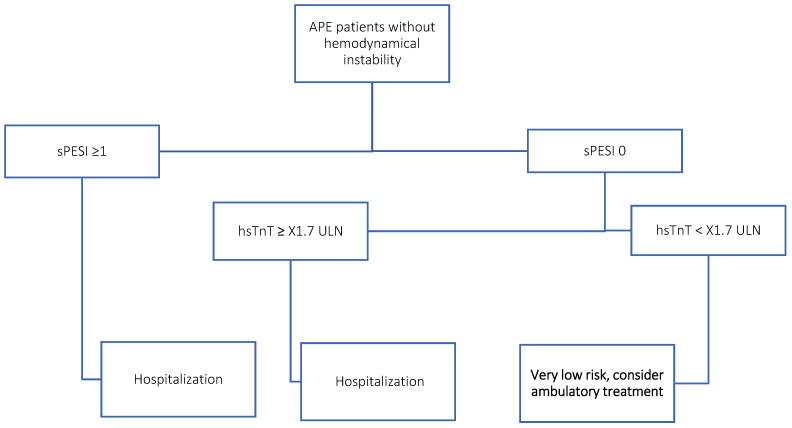
Proposed algorithm for risk stratification in normotensive patients with APE.

**Table 1 jcm-12-01276-t001:** Clinical characteristics of APE patients. Data are presented as median followed by IQR. Abbreviations: APE—acute pulmonary embolism, CE—clinical endpoint, HR—heart rate, TRPG—tricuspid regurgitation peak gradient, ULN—upper limit of normal. *p*-values calculated using the Mann–Whitney U test or Fisher’s exact test comparing CE (+) vs. CE (−) patients.

	Non-CE (N = 405)	APE-Related Mortality, Embolectomy and Thrombolysis (N = 4)	All sPESI = 0 pts (N = 409)	*p*-Value
**Female/male**	197/208	2/2	199/210	-
**Age, years**	52 (39–66)	73.5 (50–77.5)	52 (39–66)	0.12
**HR, 1 beat/min**	80 (70–90)	92 (70–105)	80 (70–90)	0.4
**Systemic systolic blood pressure, mmHg**	130 (120–140)	140 (120–155)	130 (120–140)	0.45
**Elevated troponin, n (%)**	122/405 (30%)	4/4 (100%)	126/409 (25.7%)	0.008
**Troponin/ULN**	0.2 (0–1.3)	7.8 (6.4–9.4)	0.2 (0–1.36)	0.000
**D-dimer, ng/mL**	4140 (1800–6631)	20,680 (12,580–28,780)	4169.6 (1839.5–6643.5)	0.184
**RV/LV** **4-chamber**	0.9 (0.8–1.00)	1.0 (0.8–1.6)	0.9 (0.8–1.0)	0.27
**TRPG, mmHg**	25 (20–35.5)	26.5 (17.5–38.5)	25 (20–35.5)	0.87
**RV/LV > 1.0 and/or TGPG >31 mmHg at echocardiography n (%)**	156/405 (38.5%)	4/4 (100%)	160/409 (39%)	0.02

**Table 2 jcm-12-01276-t002:** ROC analysis of troponin and RV/LV in the prediction of a complicated clinical course.

Parameter	AUC	95%CI	*p*-Value	Sensitivity	Specificity
**Troponin/ULN**	0.908	0.831–0.984	<0.001	100	79
**RV/LV > 1.0**	0.725	0.417–0.999	0.15	75	69

**Table 3 jcm-12-01276-t003:** Univariable predictors of SAE in sPESI 0 points APE patients.

Parameter	OR	95% CI	*p*
**Troponin > 1.7 ULN**	26	1.34–513	0.03
**RV/LV > 1.0**	4.6	0.41–51	0.21

**Table 4 jcm-12-01276-t004:** Predictors of study outcomes by multivariable logistic regression analysis. ULN—upper limit of normal.

Parameter	Beta-Coefficient	Standard Error	OR 95% CI	*p*-Value
**Troponin/ULN**	0.13	0.06	1.14 (1.11–1.29)	0.03
**RV/LV > 1.0**	2.48	2.24	11.9 (0.14–960)	0.26
**Age**	0.03	0.044	1.14 (0.94–1.12)	0.46

## Data Availability

Data are available from the corresponding author upon written request.

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
