# Peer review of "Plasma Troponins Identify Patients with Very Low-Risk Acute Pulmonary Embolism"

_jcm, 2023, doi:10.3390/jcm12041276_

Round 1

Reviewer 1 Report

Karolak et al. show in a prospective study (based on the PE-Aware registry) of normotensive PE patients that sPESI score of 0 alone is insufficient to detect patients at low risk of complication in case of early discharge. Indeed 22.3% of patients classified as Low risk according to sPESI have actually a troponin elevation. Authors show that troponin had the best positive predictive value for the composite endpoint (death/thrombolysis/surgical thrombectomy). This study is interesting because in clinical practice sPESI tends to be used as an ambulatory treatment decisive element but has only been validated as a marker of 30 days mortality.

Indeed, several previous studies have already shown that patients with sPESI of 0 may have other severity markers such as an elevation of the RV/LV ration or a troponin elevation.

Author Response

Dear Reviver

            Thank you for reviewing our manuscript. We are grateful for your positive opinion regarding our study.  Please find the cover letter attached to this message.

With kind regards

Reviewer 2 Report

The ESC guidelines recommend that evaluation of VD by imaging or laboratory biomarkers should be considered even in the presence of a low PESI score or a sPESI with a value of 0.

This manuscript just reinforces this idea, the algorithm proposes to identify  “very low” risk APE patients is interesting, but excludes the part with the RV dysfunction. (According to the ESC guidelines, low-risk APE is when sPESI is 0 points and RV dysfunction as assessed by echo or ct and myocardial injury markers were negative).

The most important thing is that echocardiography should be mandatory in these patients and not excluded. The number of patients is too low to confirm that echocardiography can not predicts outcomes, and this thing should be mentioned in the limitation.

The most important conclusion of this study is that highly sensitive serum troponin should be asses in each APE patient, despite the low risk. The title is too long and doesn’t imply the conclusion, which also could mislead the reader. A new title like „Troponin stratified patients with very low-risk acute pulmonary embolism” would be more suitable in my opinion.

Author Response

Dear Reviewer,

Thank you for reviewing our manuscript. According to your helpful and valuable suggestions, we proposed new, brief title: “Plasma troponins identify patients with very low risk acute pulmonary embolism”. Furthermore, we emphasized, that excluding echocardiography from risk stratification process was not our intention (page 8, lines 268-271). We added:
“Although our study did not confirm that RV/LV diameter ratio > 1.0 predicts adverse events in low risk group,  number of and endpoints is too low to exclude echocardiography from risk stratification process in those patients.”
to “Study limitations” section. In addition, paper describing role of echocardiography in predicting outcome in normotensive APE patients was cited (reference number 24; Pruszczyk P, Goliszek S, Lichodziejewska B, Kostrubiec M, CiurzyÅ„ski M, Kurnicka K, Dzikowska-Diduch O, Palczewski P, Wyzgal A. Prognostic value of echocardiography in normotensive patients with acute pulmonary embolism. JACC Cardiovasc Imaging. 2014 Jun;7(6):553-60. doi: 10.1016/j.jcmg.2013.11.004. Epub 2014 Jan 8. PMID: 24412192.)

Please find the cover letter attached to this message.

With kind regards

Reviewer 3 Report

The current manuscript titled: "Plasma troponins, not echocardiography, predict outcome in clinically stable, low risk acute pulmonary embolism patients according to sPESI score" represents an important analysis of evolving field of Cardiology and Emergency Medicine.

In my opinion, these are the adjustments which should be made to increase the value of your manuscript:

1.      The article title is not well worded, please rephrase more clearly and precisely.

2.      In Introduction chapter, please, describe the classic PE risk factors, and only then, mention COVID-19. Also, add detailed information about the pathophysiological processes linking pulmonary embolism and troponin values.

3.      Line 72: add abbreviation for “M” and “F”.

4.      The manuscript does not say anything about D-dimers, biomarker recognized as specific for PE in the ESC PE Guideline. When recommending troponins as a biomarker for PE, it is important to compare these 2 potential biomarkers, and then the results interpretation in the context of real clinical practice will become more important. It is recommended to add this data.

5.      Please indicate how the diagnosis of myocarditis was made.

6.      In Figures 1 and 3, please change “SPESI” to “sPESI”.

7.      In Table 1, please explain why “HR, 1 beat/s” was used and not “HR, beats/minute”.

8.      In logistic regression, add please Coefficient (B), SE and Exp(B)/Odds Ratio values.

9.      In the Discussion section, there is not enough comparative information with other studies.

10.  The manuscript contains some punctuation errors and typos, please revise the text (e.g., lines 33, 40, 57, 113, 125, 133, 142, 143, 166, etc.).

11.  Adapt the references according to the Journal requirements.

Author Response

Dear Reviewer,

Thank you for reviewing our manuscript. Please find answers to your valuable comments below. 

  1. The article title is not well worded, please rephrase more clearly and precisely.

    Comment:
    According to the Reviewer’s suggestion we proposed new, brief title: Plasma troponins identify patients with very low risk acute pulmonary embolism

  2. In Introduction chapter, please, describe the classic PE risk factors, and only then, mention COVID-19. Also, add detailed information about the pathophysiological processes linking pulmonary embolism and troponin values

    Comment:
    Reviewer’s suggestion is very valuable, in the “Introduction” section we added:
    1. Numerous predisposing genetic and environmental factors are known, such as lower limb fracture and joint replacement, major trauma, surgery, other causes of immobility, neoplastic diseases, estrogen use, pregnancy and others. (page 1, lines 37 – 39)
    2. Myocardial damage caused by APE depends on percentage of the pulmonary arterial bed involved. Occlusion of single segmental pulmonary artery may remain asymptomatic and clinically inconsequential. On the other hand, obstruction of pulmonary arteries by large thrombi may result in significant growth of pulmonary vessel resistance, which triggers compensatory mechanisms increasing heart rate and right ventricular (RV) contractibility in order to maintain flow through occluded vessels. As a consequence, RV oxygen demand rises, which ,in case of inadequate supply may, damage the cardiomyocytes. RV distress may be detected using imaging studies [e.g. echocardiography, computed tomography (CT)], whereas elevated serum troponin level demonstrates myocardial injury. (page 1, lines 42-50)

  3. Line 72: add abbreviation for “M” and “F”.

    Comment:
    This has been corrected (currently line 87)

  4. The manuscript does not say anything about D-dimers, biomarker recognized as specific for PE in the ESC PE  When recommending troponins as a biomarker for PE, it is important to compare these 2 potential biomarkers, and then the results interpretation in the context of real clinical practice will become more important. It is recommended to add this data.

    Comment:
    Indeed, D-dimers play important role in PE management. In the “Introduction” section we added:
    „Another biomarker specific for APE mentioned in the guidelines are d-dimers. However, unlike troponins, its clinical use is currently limited to diagnostic process rather than risk stratification” (page 1 lines 51-53).

  1. Please indicate how the diagnosis of myocarditis was made.

    Comment: In the “Results” section we wrote:
    “One patient was excluded from final analysis due to coexisting acute myocarditis, confirmed in cardiac MRI.” (page ¾, lines 144-146)

  2. In Figures 1 and 3, please change “SPESI” to “sPESI”.

    Comment: This has been corrected.

  3. In Table 1, please explain why “HR, 1 beat/s” was used and not “HR, beats/minute”.

    Comment: We wish to apologize for this error, data was actually presented in beats per minute, this was corrected (Table 1, page 4)

  4. In logistic regression, add please Coefficient (B), SE and Exp(B)/Odds Ratio values.

    Comment:
    Following the Reviewer’s comment we have added the beta coefficient and standard error values in the Table 4. (page 5 line 189)

  5. In the Discussion section, there is not enough comparative information with other studies.

    Comment:
    We wish to thank the Reviewer for that valuable suggestion, we compared our findings with the following studies addressing the issue of prognostic value of troponin assessment in low risk APE patients
    1. El-Menyar A, Sathian B, Al-Thani H. Elevated serum cardiac troponin and mortality in acute pulmonary embolism: Systematic review and meta-analysis. Respir Med. 2019 Oct;157:26-35, reference nr 22
    2. Hakemi EU, Alyousef T, Dang G, Hakmei J, Doukky R. The prognostic value of undetectable highly sensitive cardiac troponin I in patients with acute pulmonary embolism. 2015 Mar;147(3):685-694, reference nr 23

 In the „Discussion” section we added:
Subgroup analysis for low risk APE in meta-analysis by El Menyar et al. published in 2019 demonstrated connection between elevated serum troponin level and mortality for high-sensitivity (OR 6.93, 95% CI 1.34–35.78) and all troponin assays (OR 9.02, 95% CI 2.51–32.35) [22]. In the study by Hakemi et al., enrolling 298 APE low risk patients, selected according to PESI, no deaths occurred in the troponin negative group vs nine (6%) in troponin positive group. Moreover, no hard events, defined as in-hospital death, CPR or thrombolytic therapy were observed in troponin negative group vs 15 (9%), in hsTnT-positive group (p<0.001) [23]. (page 7, lines 232 – 239)

  1. The manuscript contains some punctuation errors andtypos, please revise the text (e.g., lines 33, 40, 57, 113, 125, 133, 142, 143, 166, etc.).

    Comment:
    Again, we wish to apologize for those shortcomings, text has been revised

  2. Adapt the references according to the Journal requirements.

    Comment: This has been corrected, according to the Journal requirements references are described as follows
    Author 1.; Author 2, Title of the article. Abbreviated Journal NameYearVolume, page range.

With kind regards

Round 2

Reviewer 3 Report

The Authors did not respond to all my requests. I continue to support my idea that the biomarkers investigated in this article are of no practical value without their statistical comparison with D-Dimers.

Author Response

Dear Reviewer,

Thank you for reviewing our manuscript. According to your suggestion, we included d-dimers in our analysis. We added:

  • In the “2.3 Biochemical analysis” section: “Plasma D‑dimer concentrations were measured using an automated enzyme-linked fluorescent assay VIDAS D‑Dimer Exclusion (bioMerieux, France) with a threshold of 500 ng/mL in diagnosing APE.” (lines 122-124)
  • In Table 1 now reads as follows: (page 4/5)
  • In the “Results” section: “No statistically significant difference in d-dimer level between the studied groups was noted.” (lines 170-171)
  • In the “Discussion” section: Numerous studies examined usefulness of D-dimer assessment in predicting outcome in APE, however, the results are equivocal. Meta-analysis of 22 studies by Beccattini et al. suggested association between D-dimer level and short-term and 3-month mortality in hemodynamically stable patients with APE. Nevertheless, the investigators were not able to establish an optimum prognostic cut-off value [24]. Another study by Song et al. identified D-dimer as independent predictor of in-hospital mortality (OR = 1.07; 95% CI, 1.003-1.143; P = 0.041). However, this study did not involve exclusively hemodynamically stable patients [25]. On the other hand, in the study by Stein et al., involving 292 normotensive patients, D-dimer level was not associated with increased in-hospital PE-related and all-cause mortality [26]. Current ESC guidelines do not recognize D-dimer assessment as a valuable risk stratification tool, the use of it remains limited to diagnostic process [1]. Taking available scientific data into consideration, it is possible that D-dimer might play a role in outcome prediction, however, as it may be affected by several factors (e.g. inflammatory process, pregnancy, malignancy), its use as a single predictor does not seem adequate.  (lines 247-261)

With best regards